# Effectiveness of Limestone Powder as a Partial Replacement of Cement on the Punching Shear Behavior of Normal- and High-Strength Concrete Flat Slabs

**Bilal Kamal Mohammed** [1,2,*] and **Bayan Salim Al-Numan** [3]

1   Civil Engineering Department, College of Engineering, University of Sulaimani,
    Sulaimani 46001, Kurdistan Region, Iraq
2   Surveying Department, Technical Institute of Sulaimani, Sulaimani Polytechnic University,
    Sulaimani 46001, Kurdistan Region, Iraq
3   Civil Engineering Department, Faculty of Engineering, Tishk International University,
    Erbil 44001, Kurdistan Region, Iraq; bayan.salim@tiu.edu.iq
*   Correspondence: bilal.kamal@spu.edu.iq

**Abstract:** The objective of this study is to investigate the performance of normal- and high-strength concretes including limestone powder (LP) through their mechanical properties. Moreover, sustainable flat plates made of these concretes were investigated through their punching strength. For this purpose, two different types of concrete (normal- and high-strength) with various limestone replacement ratios of 0%, 5%, 15%, and 20% by weight were designed. The fresh and hardened characteristics of the mixtures were investigated at various ages. By this means, the experimental behavior of reinforced concrete (RC) flat plate slabs made with limestone powder subjected to punching shear failure was studied. Slump value increased up to a 5% replacement of LP; after that, there was a tendency for the slump value to decrease as the replacement of limestone in normal-strength LP concrete increased. However, slump values for high-strength LP concrete increased as the LP replacement amount increased. There was a steady decrease in the compressive strength and splitting tensile strength values with the increase in LP content in normal concrete. However, in the high-strength LP concrete, with more than 10% of replacement LP, a decrease in the compressive strength values and splitting tensile strength values occurred. Compared to the control slab specimen without LP, in normal strength, the slab specimens with LP exhibit a larger ultimate shear load for slab specimens containing 5% and 10% of LP. The maximum increment for RC slabs containing 10% limestone powder was 3.8%. However, in high-strength concrete, the slab specimens with LP remained at the same ultimate shear load as control slabs, up to 10% of LP. high-strength concrete slabs with 5–20% LP showed an overall increase of (17.2%) in punching strength over the corresponding LP normal-strength concrete slabs. The corresponding increase for control slabs was 18.8%. It can be concluded that introducing LP improves the slab punching strength in a similar way that is found in non-sustainable slabs when using either normal- or high-strength concrete.

**Keywords:** mechanical properties; limestone powder; normal- and high-strength concrete; flat plate slab; punching shear

## 1. Introduction

Integrating concern for sustainability into architecture is essential to a healthier and more sustainable environment [1]. One of the many ways that engineers can implement sustainability into their work is through the materials that they use. The current conventional materials that are used in construction have limited availability, and they also create large carbon footprints, meaning that the procedures that are required to make, transport, install, and dispose of them are all very damaging to the environment and require the use of large amounts of fossil fuels and other natural resources that are being depleted and so becoming

decreasingly available [2]. Concrete is the most extensively employed construction material globally, ranking second only to water [3]. The surge in concrete demand aligns with the escalated production of its key component, cement. However, mounting concerns over the construction industry's carbon footprint and environmental repercussions underscore the imperative for innovative construction materials. This urgency becomes particularly pronounced considering the substantial energy inputs involved in manufacturing materials like cement. Notably, cement ranks as the third-largest industrial energy consumer worldwide, leaving a discernible environmental impact [4].

Recent research endeavors in cement production have pivoted towards mitigating its environmental footprint. A promising alternative explored in these efforts is limestone, which offers distinct advantages in terms of structural purity, availability, and cost-effectiveness compared to traditional cement. Consequently, many researchers have contemplated its integration as a substitute for cement. Ryno van Leeuwen conducted a comparative analysis encompassing $CO_2$ emissions, temperature effects, workability, density, shrinkage, compression strength, mortar strength, and porosity. Their findings revealed that larger proportions of added limestone exhibited notable efficacy in reducing $CO_2$ emissions. Specifically, with a 10% increment in limestone powder, emissions decreased by 8–9%. The replacement also resulted in better distribution, workability, reduced porosity, and shrinkage. However, the compressive strength decreased with an increase in limestone powder. At all levels of replacement that were tested in this investigation, the compressive strength of mortar and concrete was mostly unaffected by the particle size of the limestone that was employed [5]. Inter-ground limestone may be substituted with up to 5% of the total mass of Portland cement according to ASTM C150/C150M [6]. Calcium carbonate by mass has to comprise at least 70% natural limestone [7]. Cement may be inter-ground with or blended with limestone according to ASTM standard C595/C595M [8]. The specifications of the physical performance test are provided by ASTM standard C1157/C1157M [9]. It lists four varieties of cement: LH (low heat of hydration), MH (moderate heat of hydration), HE (high early strength), and MS (moderate sulfate resistance).

To investigate the potential impacts of varying limestone powder replacement ratios and fineness on the properties of limestone powder cement concrete, researchers undertook a comprehensive examination [10]. The findings from these experiments revealed that the incorporation of high-calcium carbonate limestone powder instigates reactions with cement, forming monocarbonate aluminates. These aluminates, in turn, serve as fillers within the concrete matrix, resulting in reduced porosity and enhanced strength. Moreover, the presence of limestone powder expedited the setting time of the concrete due to an accelerated hydration process. While particle size exhibited no significant influence on the setting time, mixes with particles of greater surface area displayed a shorter setting time, attributed to the hastened hydration process. Another property impacted by the increase in limestone powder content was the maximum heat flow, with higher values recorded as the quantity of limestone in the mixture increased. In terms of compressive strength, the specimen with 15% limestone powder had higher compressive strength due to the filler effect, accelerated hydration, and higher density than pure cement. Still, the specimens with 25% and 35% limestone powder had lower compressive strengths than pure cement because of the dilution effect, which contrasts with the filler effect. Finally, the specimens with 15% limestone exhibited smaller porosity than pure cement, and coarser limestone particles also led to a decrease in porosity and an increase in the density of concrete. In another study, Portland cement was partially replaced with limestone powders of varying particle sizes of 5, 10 and 20 μm at various replacement amounts to create Portland–limestone cement pastes. The percentages of substitution for limestone by weight were 0, 5, 7.5, 10, 12.5, 15 and 20%. Depending on compressive strength at age 1, 7, 14, 28 and 90 days and setting time, the influence of limestone powder fineness and quantity were evaluated. The observed compressive strength values were affected by the fineness of the used limestone powder. In particular, the use of limestone that was 5 μm thick seemed to provide compressive strength that is comparable to OPC control at early ages. At all ages, it appears that the filler effect is

unable to compensate for the dilution impact. According to the findings of the standard consistency tests, limestone appears to have no difference in water requirements from Portland cement. Additionally, the increase in fine particles would require a lot of water. Also, cement pastes with 5 μm of limestone exhibit a slower setting time than those with 10 and 20 μm of limestone, respectively, at the same level of replacement [11]. In a study conducted by Al-Nu'man, Bayan S., et al., the impact of incorporating limestone powder (LSP) and styrene butadiene rubber (SBR) latex into concrete mixes was examined. The concrete produced had a water/cement ratio of 0.5 and the following cementitious material ratios of sand/gravel: 1:1.8:3. LSP was utilized with cement replacements ranging from 0% to 20% by weight. For each LSP ratio, varying proportions of SBR latex—0%, 5%, 10%, and 15%—were added to strengthen the concrete matrix. Compressive and flexural strength measurements were taken at 3, 7, 14, 28, and 90 days, revealing a general improvement in both properties [12]. Another researcher explored several properties in both fresh and hardened states, along with the structural behavior of limestone powder (LP) concrete (LPC) beams. The initial investigation focused on the physical and mechanical properties of LPC with varying replacement percentages (5%, 10%, 15%, 20%, 25% and 30%). The second phase concentrated on the structural behavior of reinforced LPC beams with different reinforcement levels. The study concluded that increasing LP concentration from 5% to 30% caused a drop in compressive strength from 44.3 MPa to 24.8 MPa. With 30% LP incorporation. The 28-day splitting tensile strength was reduced by 39%, compressive strength by 71%, and flexural strength by 43% All beam specimens' flexure failed, and the ultimate load decreased by 11% when LP content was increased to 30% [13]. In another study, researchers examined the effects of partially substituting Portland cement clinker with limestone addition at different Blaine fineness values. The addition of more limestone filler led to a reduction in the compressive strength of hardened cement mortar at the same Blaine fineness. Substituting Portland cement clinker with limestone addition diluted the C3S and C2S components crucial for strength production, resulting in strength losses. The findings also indicated that adding finely ground limestone filler accelerated the rate of hydration, enhancing early-age strength development while slightly affecting consistency and setting times [14].

After a thorough examination of existing literature, it was identified that there is a research gap concerning the impact of incorporating limestone powder instead of traditional Portland cement in flat plate-reinforced concrete slabs. Specifically, there is a need to explore how this substitution influences the punching shear behavior of slabs made from normal- and high-strength concrete. Consequently, this study undertook a comprehensive experimental program to investigate the effects of using limestone powder as a partial replacement for conventional Portland cement. The research assessed various material properties such as workability, compressive strength, splitting tensile strength, modulus of elasticity, and stress–strain behavior, and then ten slab specimens were tested to investigate the punching shear behavior of both normal- and high-strength concrete flat slabs, each featuring different percentages of limestone powder and varying concrete strengths.

## 2. Experimental Program

### 2.1. Materials

This investigation utilized Ordinary Portland cement (OPC) CEM I 42.5 Type R, which is locally produced according to ASTM C109/C109M [15]. Tables 1 and 2 show the chemical composition and physical properties of OPC. Portland–limestone cement is created by substituting limestone powders (LP) for part of the Portland cement. Cement pastes with 0% to 20% by weight substitution of limestone were made. LP chemical composition is presented in Table 1. When compared to the minimum required by ASTM Standard C150/C150M [16], which is 70%, the $CaCO_3$ percentage of 99% is greater.

**Table 1.** Chemical composition of the cement and limestone powder.

| Chemical Composition | Cement Content (%) | Limestone Content% |
|---|---|---|
| $SiO_2$ | 19.17 | 1 |
| CaO | 61.23 | 51.3 |
| $Al_2O_3$ | 4.65 | 0.37 |
| $Fe_2O_3$ | 3.2 | 0.39 |
| MgO | 2.62 | 5.8 |
| $SO_3$ | 2.78 | 0.11 |
| $C_3A$ | 6.90 | - |
| L.S.F | 0.96 | - |
| $C_3S$ | 59.91 | - |
| $C_2S$ | 10.02 | - |
| $C_4AF$ | 9.73 | - |
| $Na_2O$ | - | 0.002 |
| $K_2O$ | - | 0.08 |
| MN | - | 0.01 |
| $P_2O_5$ | - | 0.05 |
| $CaCo_3$ | - | 99 |
| Ti | - | 0.00003 |
| Loss of ignition | 3.60 | - |
| Insoluble residue | 0.32 | - |

**Table 2.** Physical properties of the cement.

| Physical Test | Result | Limits of Iraqi Specification No.5/1984 |
|---|---|---|
| Compressive strength $kg/cm^2$ | | |
| For 2 days | 360 | lower limit 20 $kg/cm^2$ |
| For 28 days | 498 | lower limit 425 $kg/cm^2$ |
| Time of setting | | |
| Initial setting time | 125 min | not less than 45 min |
| Final setting time | 205 min | not more than 600 min |
| Blaine ($cm^2/g$) | 3065 | 2300 $cm^2/g$ |
| Expansion (mm) | 1 | not more than 10 mm |

Crushed limestone aggregate, with a maximum size of 12.5 mm, was the natural coarse aggregate that was locally accessible. The fine aggregate was natural sand with a higher fineness modulus of 3.07, satisfying ASTM specification limits. There was a lot of discussion and analysis of various fields and data [17]. The physical properties of coarse and fine aggregates are shown in Table 3. Sieve analysis on both coarse and fine aggregates and the lower and upper limitations set by ASTM C33 [18] are shown in Figure 1. In HSC mixes, a superplasticizer was used to reduce the w/c ratio. To reduce the water-to-cement ratio, HSC mixes were treated with a superplasticizer. PASS 450 was available locally and met the requirements of ASTM C494 [19]. The concrete was mixed and cured using water from the tap.

**Table 3.** Physical properties of fine and coarse aggregate.

| Properties of Aggregate | Fine Aggregate | Coarse Aggregate |
|---|---|---|
| Percentage of water absorption | 2.46% | 1.18% |
| Bulk specific gravity, Dry | 2.6 | 2.62 |
| Bulk specific gravity, SSD | 2.66 | 2.65 |
| Apparent specific gravity | 2.77 | 2.70 |
| Uncompacted bulk density kg/m$^3$ | 1698 | 1429 |
| Compacted bulk density kg/m$^3$ | 1848 | 1558 |
| Loss Angeles abrasion | - | 8.55% |
| Finesse modules ASTM C136 [20] | 3.07 | 2.89 |

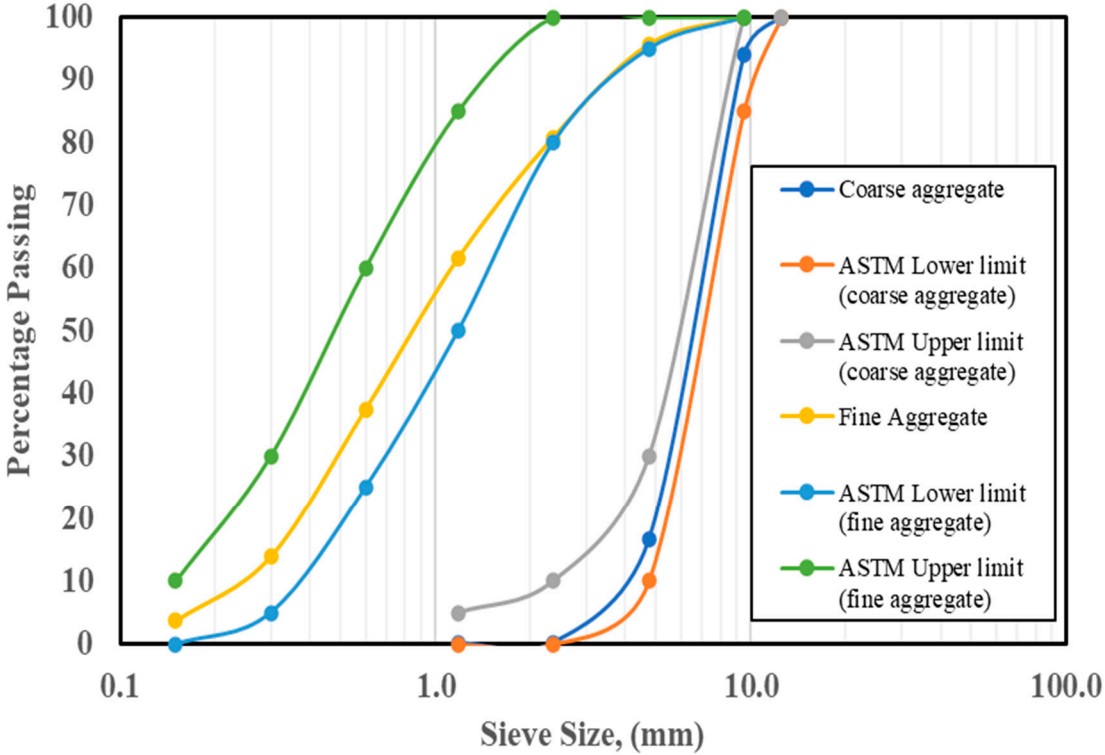

**Figure 1.** Particle size distribution for fine and coarse aggregates.

*2.2. Properties of Reinforcing Steel Bars*

Two-way flexural reinforcement was installed in each slab, consisting of Ø6 mm bars that were deformed and positioned in the tension face with a clear cover 8 mm below the mesh. The characteristics of the Ø6 mm bars that were examined are shown in Table 4.

**Table 4.** Properties of reinforcing steel bars.

| Diameter (mm) | Yield Point (MPa) | Tensile Strength (MPa) | Elongation% | Weight for kg/mL | Elastic Modulus (GPa) |
|---|---|---|---|---|---|
| 5.91 | 541 | 653 | 5.73 | 0.217 | 177.24 |

*2.3. Specimen Details and Test Setup*

This study aimed to compare the punching shear behavior of RC slabs with varying levels of limestone powder replacement. Ten specimens with test variables of limestone powder replacements and the cylinders for compressive strength, tensile strength, and modulus of elasticity of concrete for each specimen were manufactured.

The strength of the concrete was designed to be 38 MPa and 85 MPa. In the names of specimens, N-Co and H-Co denote specimens that only used OPC with normal- and high-strength concrete, respectively; NCL denotes a specimen that has normal strength with limestone powder; HCL denotes a specimen that has high strength with limestone powder; 5, 10, 15 and 20% denote the replacement level of limestone powder used in the concrete.

The slab specimens were reinforced with fifteen deformed steel bars of 6mm diameter in each direction, i.e., $\rho$ = 1.55%. The column stubs were reinforced with eight Ø6 mm deformed steel bars cut to a length fifteen millimeters below the column's height and two Ø6 mm deformed steel stirrups spaced fifty millimeters apart. All of the specimen and reinforcement dimensions are provided in Figure 2.

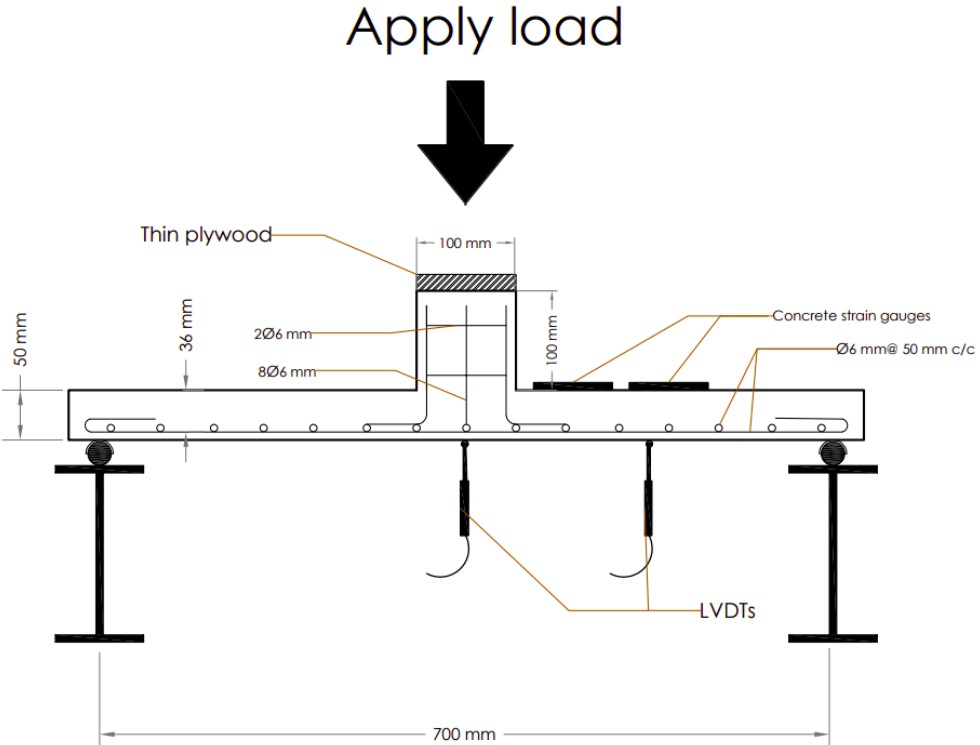

**Figure 2.** Dimensions and reinforcement of the slab specimen.

To measure the deflection, the three linear variable displacement transducers (LVDT) were set up: one at the bottom of the center of the span of the specimen and two others at the bottom of the quarter of the specimens. Additionally, the stress and strain states of the concrete strain gauge for concrete were installed at distances of d/2 and 3d from the face of the column of the slab specimens, as shown in Figure 2.

The cement/sand/coarse aggregate/water proportions are presented in Table 5. For each RC slab specimen, ten 100 mm × 200 mm concrete cylinders were cast. The compressive strength, splitting tensile strength, modulus of elasticity, and Poisson's ratio of the concrete with and without limestone powder utilized in this study were determined by casting the concrete on these specimens, as shown in Table 6. The concrete specimens were left in the controlled conditions of the laboratory for a full day after casting. After that, the concrete samples were removed from their molds and placed in water at a temperature of 23 ± 0.5 °C for 7, 28 and 56 days.

**Table 5.** Mix proportions of LP concrete.

|  | **Mix Proportion** | **W/C** | **Superplasticizer** | **Cement (kg/1 m³)** |
|---|---|---|---|---|
| Normal strength | 1:3.58:2.75 | 0.67 | - | 300 |
| High strength | 1:2.42:2.214 | 0.407 | 0.8% | 400 |

**Table 6.** Test results of LP and control concretes.

| Mix Symbol | Slump (mm) | Compressive Strength (MPa) | | | $fc'7/fc'28$ | $fc'28/fc'56$ | $E_{C,exp}$ (GPa) | $\mu_{C,exp}$ |
|---|---|---|---|---|---|---|---|---|
| | | **7 Days** | **28 Days** | **56 Days** | | | | |
| N-Co | 160 | 24.56 | 33.02 | 38.52 | 0.74 | 0.86 | 23.8 | 0.130 |
| H-Co | 175 | 65.55 | 83.1 | 87.36 | 0.79 | 0.95 | 33.9 | 0.209 |
| NCL5 | 175 | 21.32 | 32.72 | 35.13 | 0.65 | 0.93 | 23.9 | 0.131 |
| NCL10 | 140 | 22.11 | 31.8 | 32.53 | 0.70 | 0.98 | 21.4 | 0.122 |
| NCL15 | 130 | 19.22 | 28.82 | 30.02 | 0.67 | 0.96 | 19.9 | 0.141 |
| NCL20 | 120 | 16.14 | 23.78 | 25.94 | 0.68 | 0.92 | 19.6 | 0.138 |
| HCL5 | 190 | 73.49 | 86.45 | 91.43 | 0.85 | 0.95 | 36.8 | 0.256 |
| HCL10 | 200 | 69.18 | 84.95 | 87.21 | 0.81 | 0.97 | 32.8 | 0.197 |
| HCL15 | 210 | 67.76 | 78.22 | 82.78 | 0.87 | 0.94 | 32.6 | 0.174 |
| HCL20 | 210 | 61.02 | 71.44 | 75.48 | 0.85 | 0.95 | 29.6 | 0.165 |

## 3. Test Results

### 3.1. Slump Test (ASTM C143) [21]

The results of the slump tests on the control and LP concrete specimens are shown in Table 6. The test results showed that the control mix for normal- and high-strength concrete had approximately the same workability.

In the case of high-strength limestone concrete (HCL) mixes, it was observed that LP significantly enhanced workability when increasing the LP replacement ratio compared to the control mix; the maximum slump was 210 mm for a 20% replacement ratio. For normal-strength limestone concrete (NCL) mixes, workability increased to 175 mm for a concrete mix containing 5% LP and then decreased gradually to 120 mm for concrete containing 20% LP. The combination of superplasticizer and high cement content results in an extremely low w/c for high-strength concrete. Due to insufficient space to accommodate the reaction products, the cement in this concrete cannot completely hydrate [22]. In this case, replacing cement with limestone can change the way cement particles are packed and raise the Portland clinker's hydration level, both of which improve strength.

### 3.2. Compressive Strength of LP Concretes

Figure 3 illustrates the effect on the compressive strength of partially replacing limestone powder at different percentages. The compressive strength value of the normal-strength control specimen at day 56 is 38.5 MPa, while the compressive strength values at day 56 for normal-strength LP concrete specimens are shown to be 35.1, 32.5, 30.0 and 25.9 MPa as the LP substitutes' percent of cement is increased by 5%, 10%, 15% and 20%, respectively, by weight. The percentages of strength reduction are 8.8%, 15.6%, 22.1% and 32.7% when compared to the control specimen.

However, for high-strength LP concrete, the compressive strength was increased by about 4.6% (from 87.36 to 91.43 MPa) when 5% of cement was replaced by limestone powder. At 10% replacement of cement, the LP concrete had the same compressive strength as the control specimen, which was 87.2 MPa. However, when the LP replacement percentage of cement was increased by 15% and 20%, the compressive strength was reduced, and the percentage of compressive strength reduction was 5.3% and 13.6%, respectively. For both

normal- and high-strength LP concretes (except for the HCL5 mixture), this decrease in the compressive strength is related to a dilution effect that reduces the hydration of cement. This observation supports the conclusions made by [10,11].

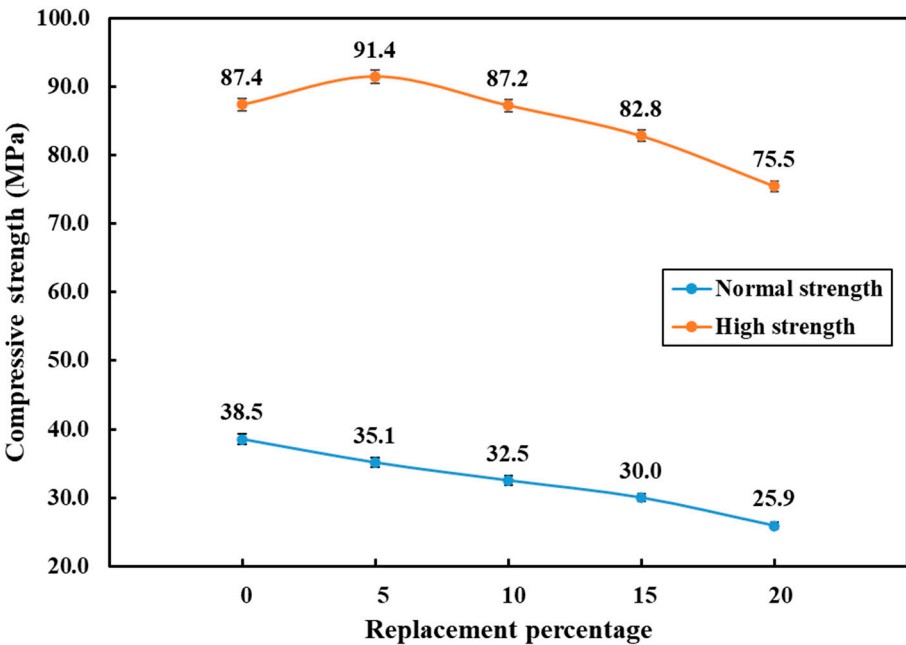

**Figure 3.** Effect of LP with various percentage replacements on the compressive strength after 56 days.

From Table 6, the 7- and 28-day strength ratio [$fc'$ (7 d)/$fc'$ (28 d)] shows approximately 6% change in compressive strength as the LP percentage was increased from 5% to 20%. For normal concrete, it was between 30 and 35%, but for high-strength concrete it was between 13 and 19%, while the 28- and 56-day strength ratio [$fc'$ (28 d)/$fc'$ (56 d)] shows less change. For normal concrete, it was between 2 and 8%, but for high-strength concrete, it was between 3 and 7%. The failure shapes for all samples after the compressive strength test are shown in Figure 4.

### 3.3. Splitting Tensile Strength (ASTM C496) [23]

The results of tests performed to establish the specimens' splitting tensile strength are represented in Figure 5. In general, the results demonstrated that using limestone powder for high-strength concrete enhanced the splitting tensile strength of concrete up to a specific replacement ratio. The maximum splitting tensile strength is obtained at a range of optimum limestone powder replacement levels from 5% to 10% for high-strength concrete.

In high-strength LP concretes, splitting tensile strength was increased by 14.5% when 5% of cement was partially replaced using LP, and when 10% of cement was partially replaced, they had nearly the same splitting tensile strength as the control concrete mixture. However, the tensile strength was decreased by about 7.3% and 10.9% when the replacement ratio was 15% and 20%, respectively, as shown in Figure 5.

The control mixtures' 56-day strength in normal-strength concretes is 4.3 MPa; however, when LP is added, the strength begins to decrease to 4.2, 3.7, 3.5 and 3.1 MPa as the LP replacement percentage of cement increases by 5, 10, 15 and 20%, respectively. For a 5–20% LP percentage, the percentage of strength decrease varied from 2.3% to 27.9%.

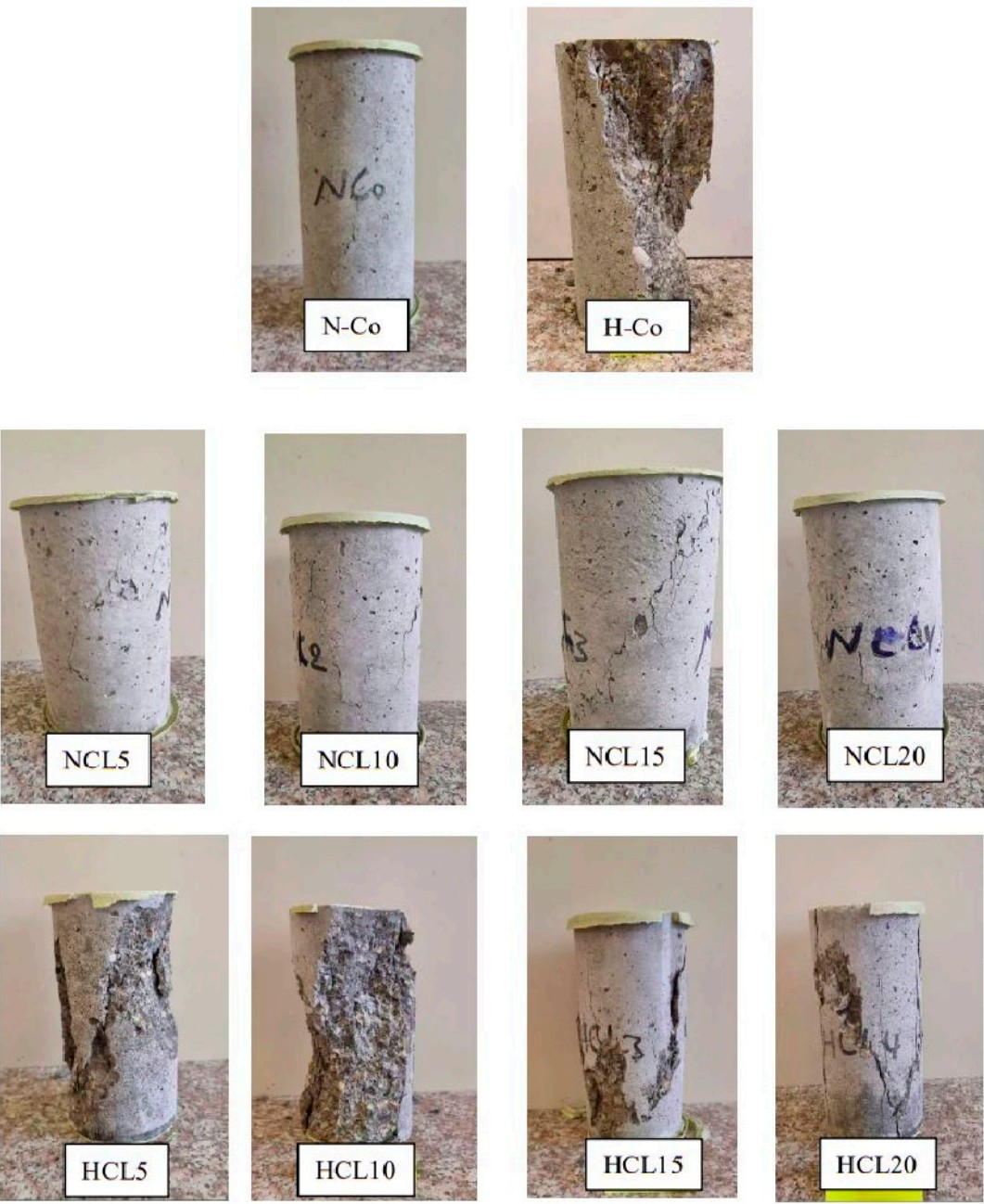

**Figure 4.** Failure modes of cylindrical specimens at 56 days compression test.

However, for normal-strength concrete, when using incrementally increasing amount of limestone powder, the value of splitting tensile strength is decreased. This reduction in splitting tensile strength can be attributed to a decrease in the amount of hydration products, which primarily results in larger porosity, a weaker interfacial transition zone, and reduced bonding strength [24].

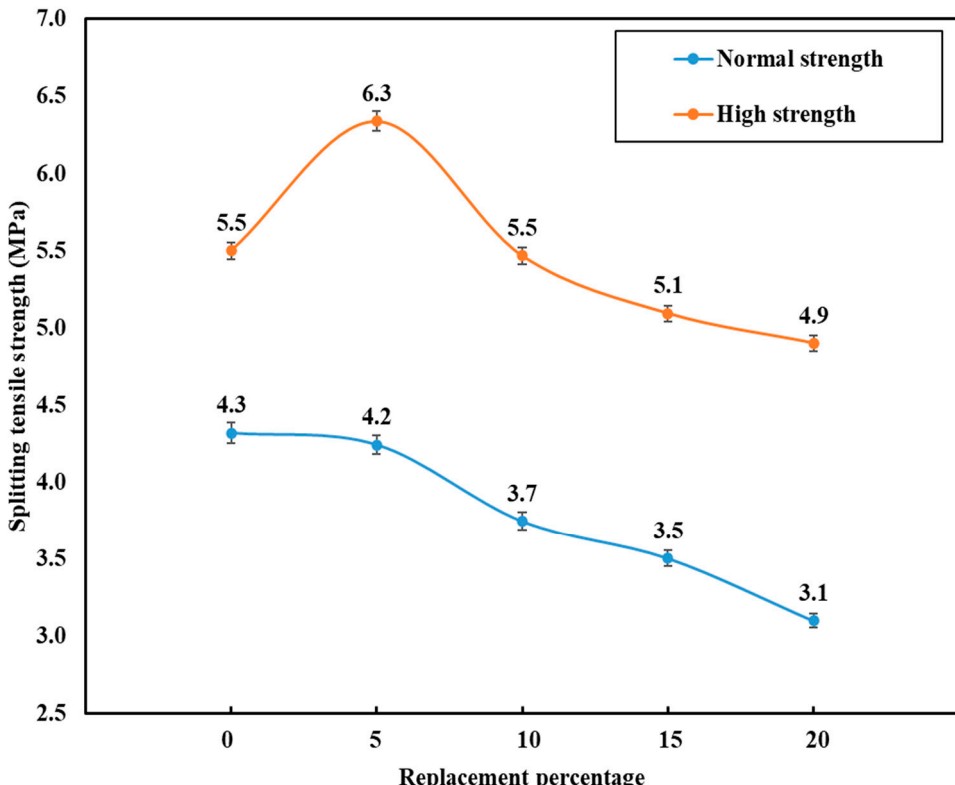

**Figure 5.** Effect of LP with various percentage replacements on splitting tensile strength after 56 days.

### 3.4. Modulus of Elasticity of LP Concretes

The modulus of elasticity and Poisson's ratio were determined based on the test procedure described in ASTM C469 [25]. The results are presented in Figure 6 and Table 6. There is a slight decrease in elastic modulus for normal-strength LP concrete mixes compared to the control mix. The reductions in elastic modulus of concrete were 10.1, 16.4 and 17.6% for the LP contents of 10, 15 and 20%, respectively. However, a mixture with 5% LP had an elastic modulus that is nearly comparable to the control mixture. On the other hand, the optimum LP content for high-strength LP concrete mixes was found to be 5%, and the concrete mixture with 5% LP had an 8.6% greater elastic modulus than the reference mixture. Moreover, when LP content increased, there was a slight decrease in elastic modulus compared to the control mixes. The reduction was 3.2, 3.8 and 12.7% for 10, 15 and 20% LP contents, respectively. The Poisson's ratios, as shown in Table 6, ranged from 0.122 to 0.148 for normal-strength LP concrete mixes, while for high-strength LP concrete, it ranged between 0.165 and 0.256. Similar to the modulus of elasticity, there is a slight variation in those values obtained for LP concrete compared to normal concrete. According to research by M. S. Meddah et al. [26], the diluting effect of LS can cause a reduction in the elastic modulus of concrete when it is used to replace cement.

### 3.5. Compressive Stress–Strain Behavior Relationship

Both axial compressive and radial tensile strains were recorded simultaneously, corresponding to the compressive stress up to the peak compressive loads. The stress–strain curves show the nonlinear characteristics of the concrete. The stress–strain curves were linear initially and then smoothly changed direction when stiffness was decreased until the maximum value of compressive strain was reached. There was no descending part in the stress–strain curves for any of the concrete specimens because of their brittle nature.

When comparing the stress values of the mix NCL5 to those of the control mixture, there was a relative improvement in the axial strain of NCL5, as seen in Figure 7. While mixes containing 10% and 15% LP show the same reduction when compared to the control

mixture, the mixture containing 10% LP has stress and corresponding strain values that are similar to those of the mixture containing 15% LP. Less reduction can be seen for mixtures with 20% LP than those containing 10% and 15% LP. In terms of radial strains of NCL mixes, all the NCL mixtures' stress–strain curves are smaller than those of the control mixture, and as the LP replacement level increases, the concrete's stiffness reduces proportionately. Regarding HCL Figure 8, all mixtures (with or without LP) typically exhibit comparable behaviors in their stress–strain curves. In comparison to the control mixture, the stress values for the same axial strain levels of the HCL5 mixture show a relative improvement. It was found that the stiffness of concrete specimens with 10% and 15% LP was nearly the same as that of the control mixture.

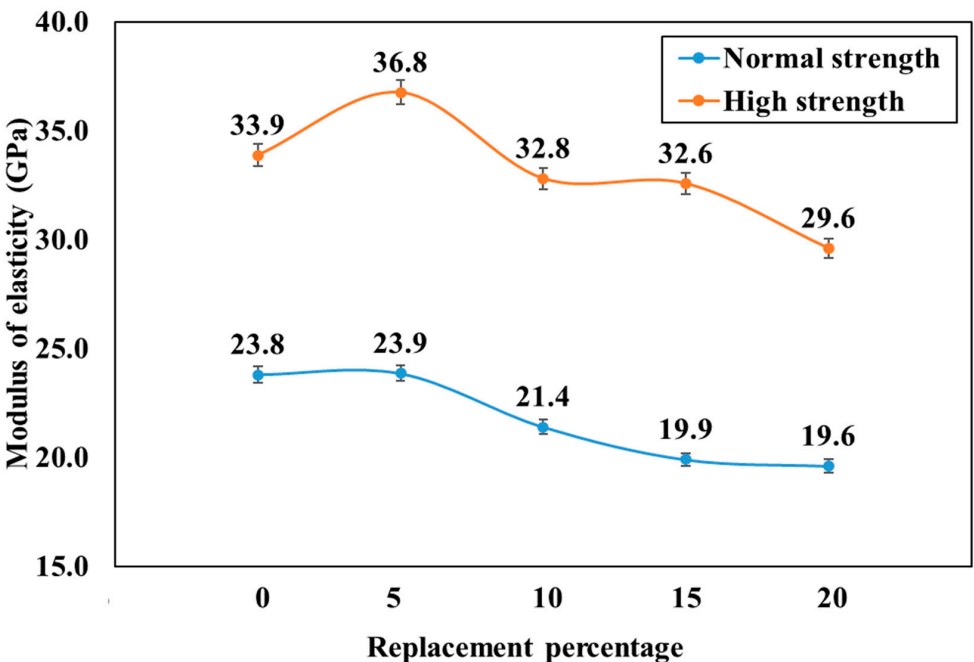

**Figure 6.** Effect of LP with various percentage replacements on elastic modulus.

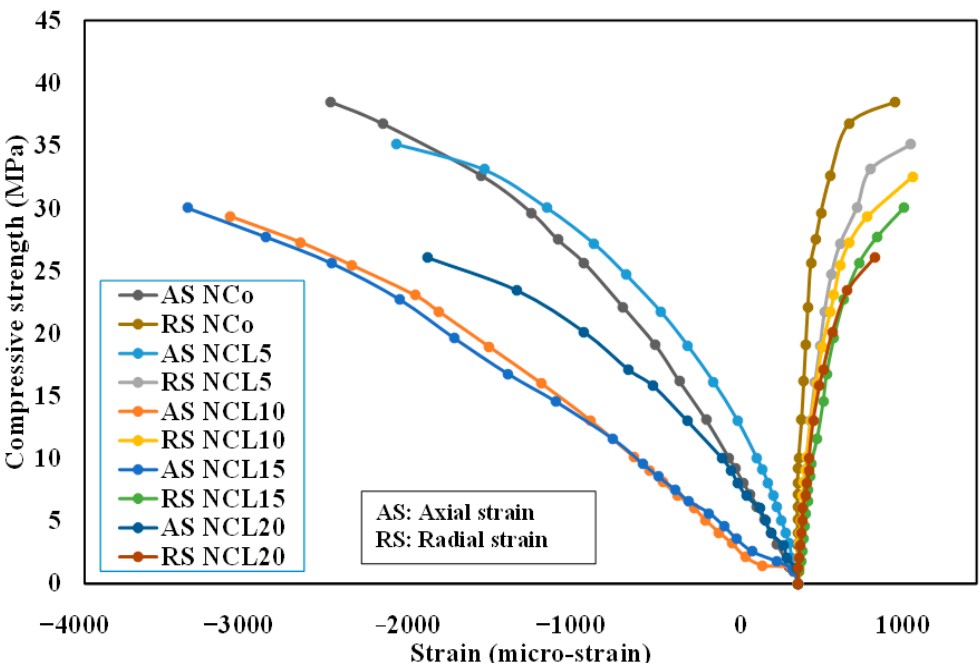

**Figure 7.** Compressive stress–strain behavior for NCL samples compared with control mix.

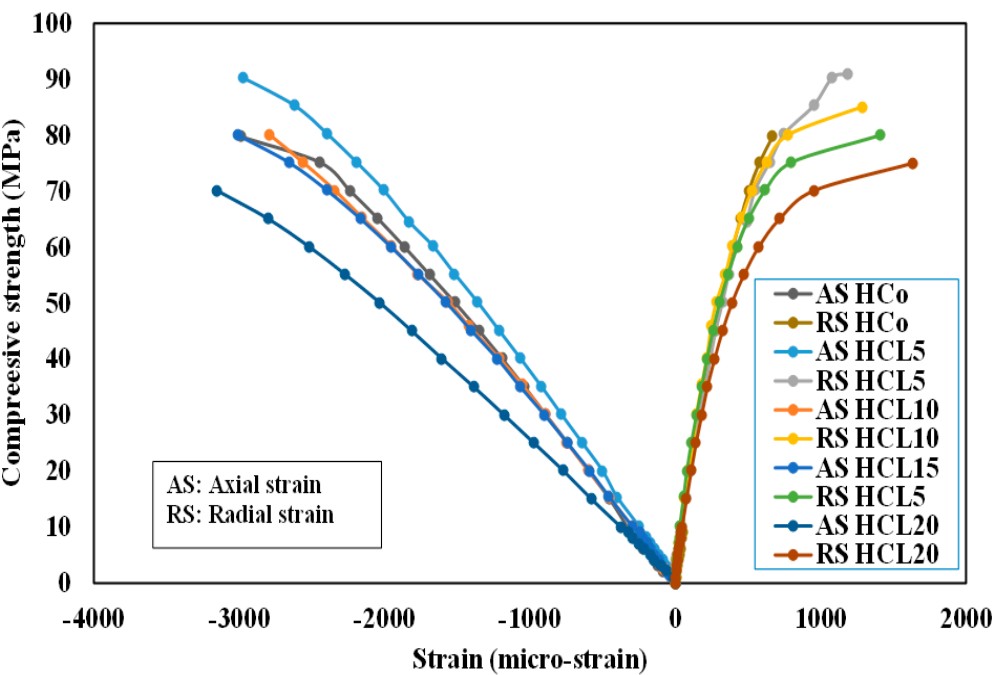

**Figure 8.** Compressive stress–strain behavior for HCL samples compared with control mix.

*3.6. Punching Load Capacity*

Figure 9 shows the ultimate failure load of control and LP concrete flat slab specimens. The punching load value of normal-strength concrete control slab specimens at 56 days is 63.9 kN, while the punching load values at 56 days for normal-strength LP concrete slab specimens are (65.0, 66.3, 62.5 and 61.1) kN as the LP replacement amount of cement is raised by 5%, 10%, 15% and 20%, respectively. Compared to control specimens, the percent of punching load was increased by 1.7 and 3.8%, respectively, when 5% and 10% limestone powder was partially replaced. However, the ultimate punching load decreased by about 2.2 and 4.4% when the replacement ratios were 15% and 20%, respectively.

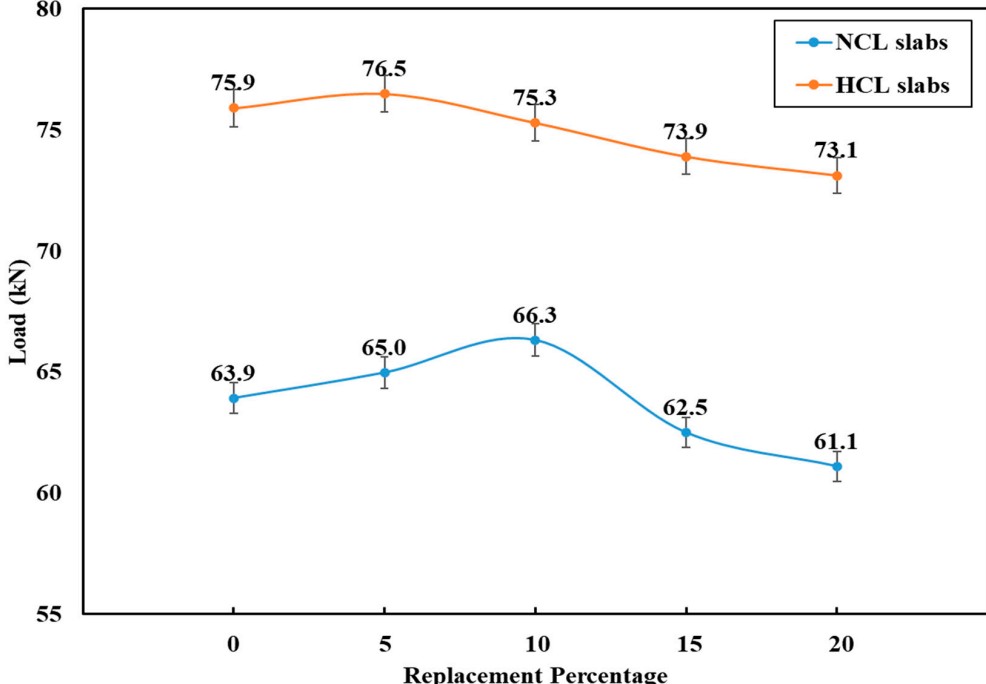

**Figure 9.** Effect of LP with various percentage replacements on punching load for slab specimens.

In high-strength concrete slabs, punching load capacity was increased slightly by 0.8% when 5% of limestone powder was partially replaced. At 10% limestone powder, slabs had nearly the same punching load capacity as the control slab specimen, while the punching load decreased by about 2.7 and 3.7% when the replacement ratio increased to 15% and 20%, respectively.

The punching strength of 5–20 percent-LP high-strength concrete slabs is overall 17.2% higher than that of corresponding LP normal-strength concrete slabs. For control slabs, the comparable increase is 18.8%. It is concluded that the utilization of LP increases the punching strength of the slab in a way comparable to that of non-sustainable slabs made of high-strength concrete.

### 3.7. Load-Deflection Behavior

The variation of the deflections during the punching test of the flat slab specimens is presented in Figures 10–13 in the mid-span and quarter-span of the flat slab specimens, where the quarter value is the average value of the two LVDTs. The nearly linear load-deflection relationship is detected, and the deflection is almost insignificant as the slabs preserve a relatively high stiffness before concrete section cracking. After the first crack appears, the deflection of flat slabs largely depends on the concrete's compressive strength and the percentage replacement of LP.

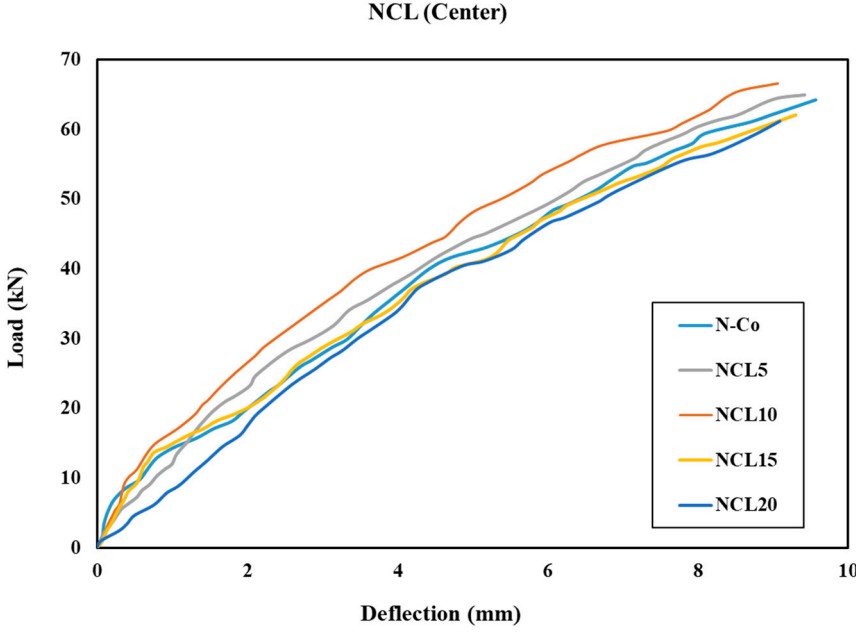

**Figure 10.** Load–center deflection curve of the NCL slab specimens.

The load-deflection for control slab specimen N-Co is shown in Figures 10 and 11 to compare the results to the slab specimens NCL5, NCL10, NCL15 and NCL20, whereas LP contents differed. When slabs fail due to pure punching failure, these linear load deflections are predictable. The actual flexural bending moments in the slabs are consequently smaller than the maximum bending capacity because of the sufficient reinforcing ratio, $\rho = 1.55\%$.

Compared to the load-central defection curves obtained for NCL5 and NCL10 with a 5 and 10% LP content, which attained their optimum value at 10% LP, the slab specimens N-Co show a higher deflection and a lower ultimate load. However, compared to the control slab specimen, the loading response for slabs NCL15 and NCL20 with 15% and 20% LP nearly follows the same pattern and has a higher deflection with a slightly lower load. With a few variations in the extent of deflections, the load-defection curves at the quarter-span and mid-span exhibit comparable behaviors. Compared to other slab specimens, the slab NCL15 exhibits greater deflection at low loads. This could be the result of an early radial

crack that ran from the column's center to the slab edge. An additional explanation is the decreased elastic modulus with large doses of LP.

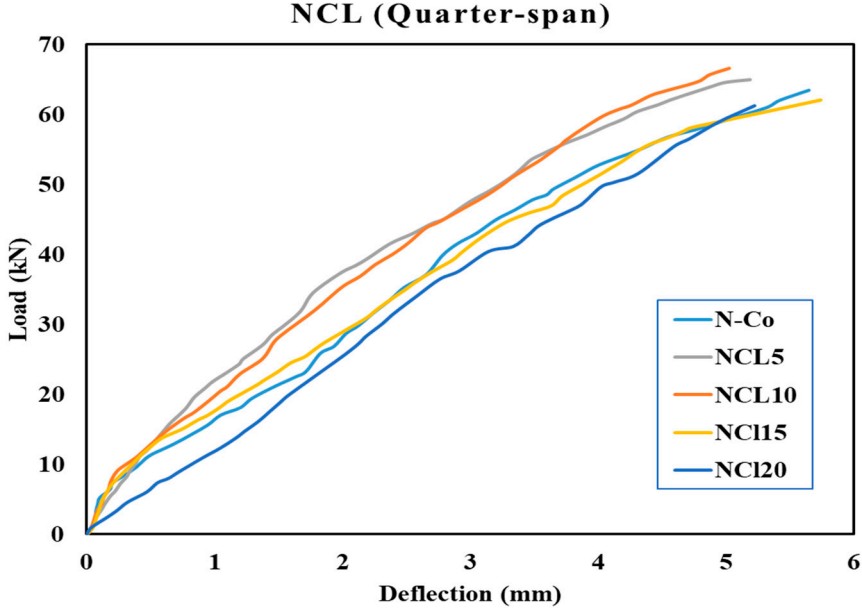

**Figure 11.** Load–quarter span deflection curve of the NCL slab specimens.

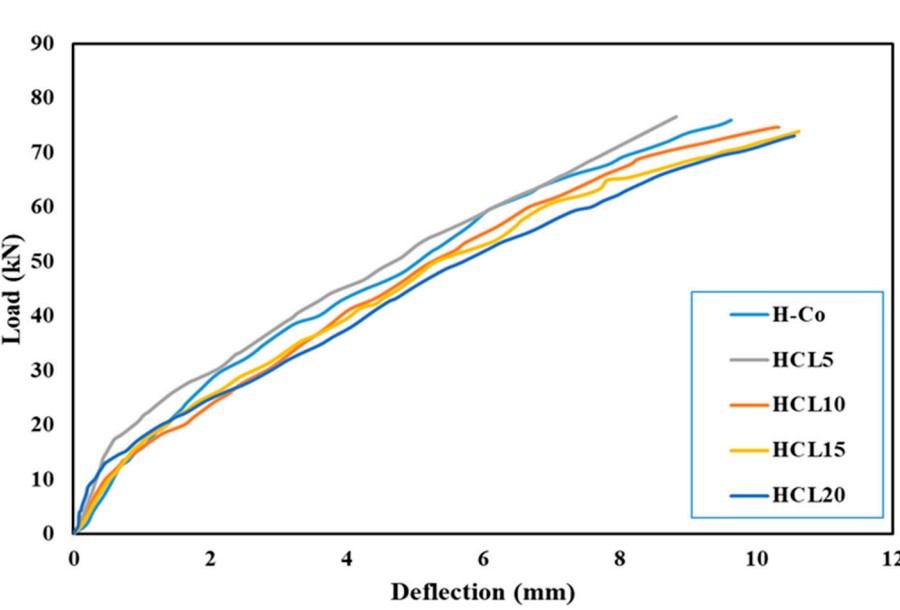

**Figure 12.** Load–center deflection curve of the HCL slab specimens.

Table 7 overviews the first crack and punching failure load and corresponding deflections at the center and quarter-span for all CL slab specimens. The first cracks appeared at loads 11.5, 11.7, 11.4, 10.4 and 8.2 kN for the slabs N-Co, NCL5, NCL10, NCL15 and NCL20, respectively. At first crack loads, the center deflection is small, measuring between 0.54 and 0.98 mm. When comparing the NCL5 and NCL10 slabs to the control slab specimen, the first crack load findings indicate nearly the same values; however, when comparing the NCL15 and NCL20 slabs to the control slab N-Co, the values are approximately 11 and 40% lower, respectively.

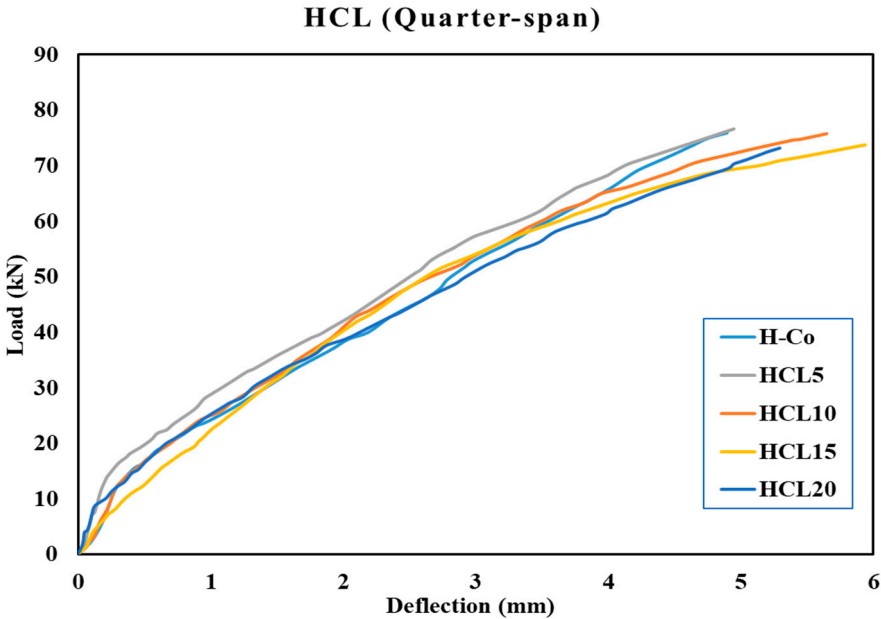

**Figure 13.** Load–quarter span deflection curve of the HCL slab specimens.

**Table 7.** Load and deflections at first crack and failure for Group CL.

| Group | Slabs | Stage | Load (kN) | Deflection at Center-Span (mm) | Deflection at Quarter-Span (mm) |
|---|---|---|---|---|---|
| Control | N-Co | 1st crack | 11.5 | 0.69 | 0.52 |
| | | ultimate | 63.92 | 9.49 | 5.68 |
| NCL | NCL5 | 1st crack | 11.7 | 0.96 | 0.45 |
| | | ultimate | 64.97 | 9.42 | 5.19 |
| | NCL10 | 1st crack | 11.4 | 0.54 | 0.43 |
| | | ultimate | 66.32 | 9.02 | 4.98 |
| | NCL15 | 1st crack | 10.4 | 0.58 | 0.37 |
| | | ultimate | 62.5 | 9.38 | 5.82 |
| | NCL20 | 1st crack | 8.2 | 0.98 | 0.65 |
| | | ultimate | 61.1 | 9.05 | 5.20 |
| Control | H-Co | 1st crack | 14.8 | 0.90 | 0.395 |
| | | ultimate | 75.9 | 9.63 | 4.90 |
| HCL | HCL5 | 1st crack | 16.3 | 1.06 | 0.31 |
| | | ultimate | 76.5 | 8.83 | 4.94 |
| | HCL10 | 1st crack | 14.6 | 0.91 | 0.40 |
| | | ultimate | 75.3 | 10.32 | 5.60 |
| | HCL15 | 1st crack | 15 | 0.85 | 0.61 |
| | | ultimate | 73.9 | 10.63 | 5.96 |
| | HCL20 | 1st crack | 14.5 | 0.66 | 0.40 |
| | | ultimate | 73.12 | 10.56 | 5.29 |

Figures 12 and 13 show the load-deflection behaviors at the center of the span and the quarter-span for slab group HCL with the H-Co slab specimen. At all loading stages, at the same load values, all central deflection curves for HCL slabs are lower than NCL deflections.

The final deflections at the quarter-span of the specimens are over 53% less than the deflections measured at the center span of the slab specimens, with comparable trends in the load-defection curves at the quarter-span and mid-span. Slab specimens H-Co with a 0% LP content demonstrate a higher deflection and a slightly lower ultimate load on the load-central defection curve than HCL5 with a 5% LP content, which obtained the optimal value. Slab HCL10, HCL15 and HCL20 with 10%, 15% and 20% LP, on the other hand, showed a loading response with higher deflections and a lower load than the control slab specimen; among the HCL slab specimens, HCL20 had the lowest ultimate load value and HCL15 the highest deflection value.

For each CL slab, Table 7 displays the first crack and punching failure load together with the associated defections at the center and quarter-span. The first cracks appeared at loads 14.8, 16.3, 14.6, 15 and 14.5 kN for the slabs H-Co, HCL5, HCL10, HCL15 and HCL20, respectively. The low range of the center deflection at first crack loads is 1.06 mm to 0.66 mm. The first crack load measurements for HCL5 are approximately 10% greater than those for the control slab H-Co, but other HCL slab specimens have nearly the same load value as the first crack.

*3.8. Load–Concrete Strains Relationship*

Concrete strain gauges were installed at several identical points on the slabs' top compression face to measure the strain variations associated with loading. The data logger collected all the data from strain gauges and load cell every second. Four concrete strain gauges with a length of 80 mm were positioned at four different distances from the column face to record the compressive strains applied to the slabs' top compression face. Two strain gauges were placed half the effective depth (0.5 d) away from the column face, while the other two were placed three times the effective depth (3 d) away, as shown in Figure 14. The relationship between load and strain in concrete is shown in Figures 15–24.

Generally, for all slabs, the following points are of interest:

- In most slabs, it can be seen that the curves change in slope at a specific point, which can be considered as the first cracking load of the slab specimens [27].
- The maximum compressive strain in all normal-strength concrete slab specimens was in slab (NCL20) which reached a value of 0.002302 (which is significantly less than the ultimate concrete strain of 0.003 specified by the ACI-code, indicating that slabs are prone to punching critical failure), representing 76.7% of the ultimate strain (0.003). The maximum compressive strain in all high-strength concrete slab specimens was in slab (HCL15) which reached a value of 0.001485, representing 49.5% of the ultimate concrete strain.
- When the load is up to 61% to 95% of the punching failure load, the curves illustrate the case of gradually decreasing compressive strain on concrete leading toward failure. These findings match with punching tests using flat RC slabs reported by other authors [28,29]. Tensile strains can be noticed shortly before punching. This occurrence can be clarified by the formation of an elbow-shaped strut with a horizontal tensile member [29].
- From Table 8, the ultimate concrete strain of NCL slabs measured was between 1448 and 2302 micro-strain (0.001448–0.002302) and between 1371 and 1485 micro-strain (0.001371–0.001485) for HCL slabs. This demonstrates that the use of LP increases the maximum compressive strain of CL slabs.
- The strain gauges at half the effective depth (d/2) recorded greater values than the ones at three times the effective depth (3 d). The ultimate strain gauge values at locations 0.5 d from the column's face are (3–5) times those for strain gauges at 3 d from the column's face.

- Figures 15–24 show that compressive strains increased when compressive strength was decreased for the same replacement ratio of the slab specimens.
- It may be noticed that the LP concretes, in general, yield typical load–strain relationships to corresponding control ones. The load–strain curve was linear until the first cracking. There was a significant decrease in slab stiffness throughout this occurrence. As the load increased, more cracks developed, some of which expanded in radial directions. Just before punching failure, the load–strain curves achieved their maximum strain values, which were followed by an unexpected decline in strain. Punching shear failure was noticeable on the slab surface at this stage.

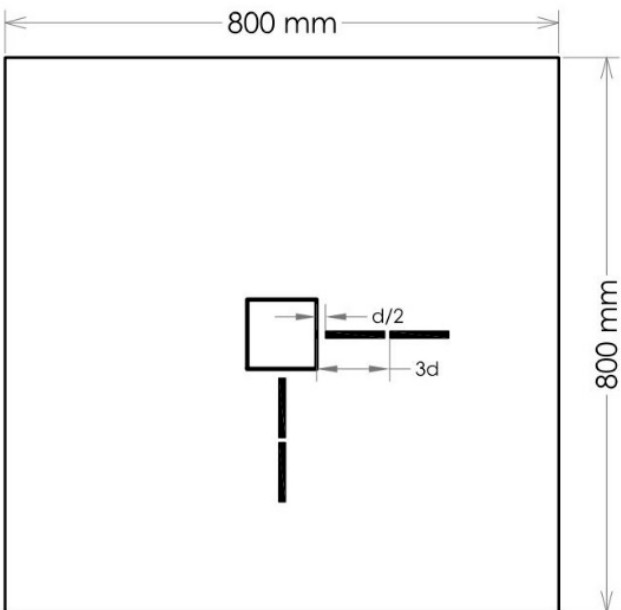

**Figure 14.** The position of concrete strain gauges.

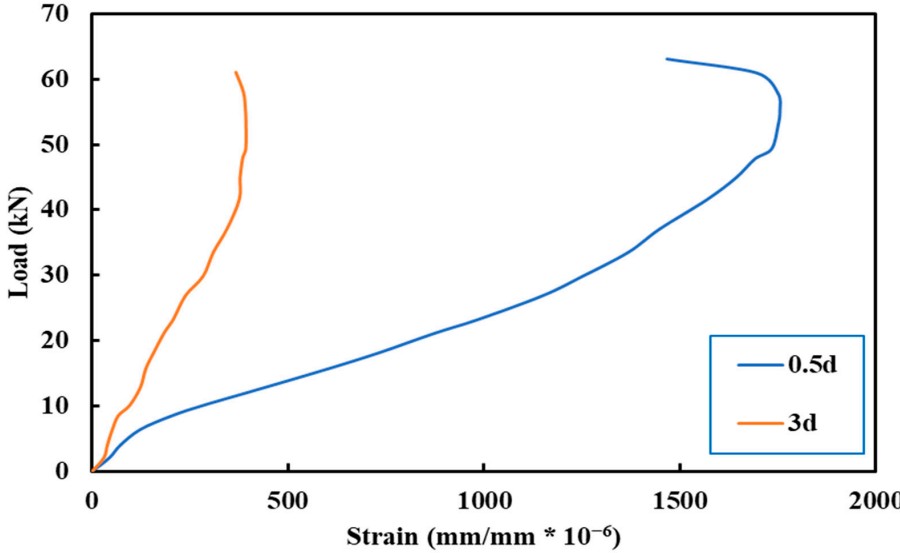

**Figure 15.** Load versus 0.5 d and 3 d concrete strains for N-Co slab specimen.

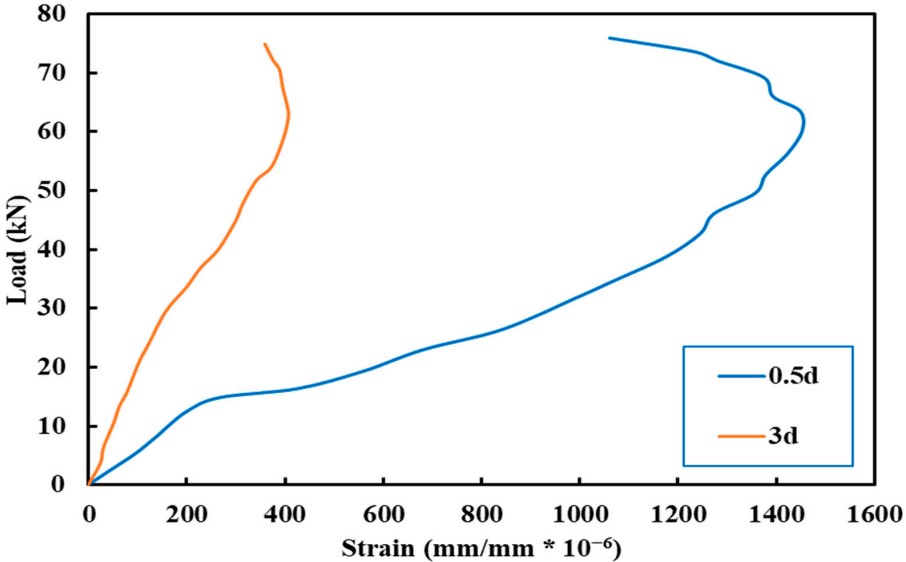

**Figure 16.** Load versus 0.5 d and 3 d concrete strains for H-Co slab specimen.

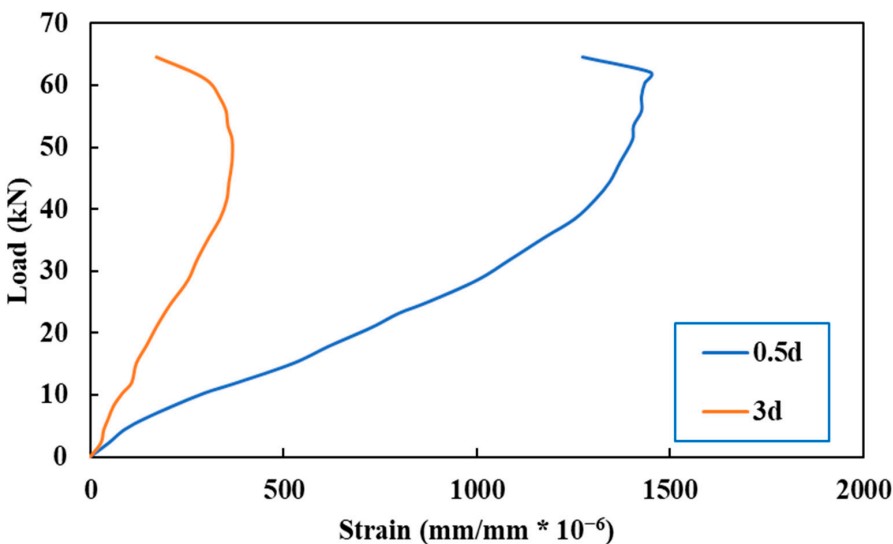

**Figure 17.** Load versus 0.5 d and 3 d concrete strains for NCL5 slab specimen.

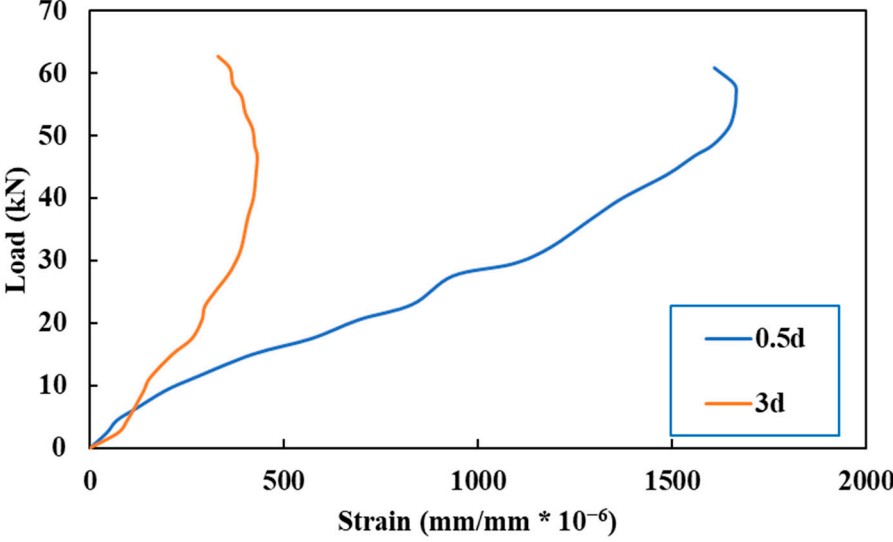

**Figure 18.** Load versus 0.5 d and 3 d concrete strains for NCL10 slab specimen.

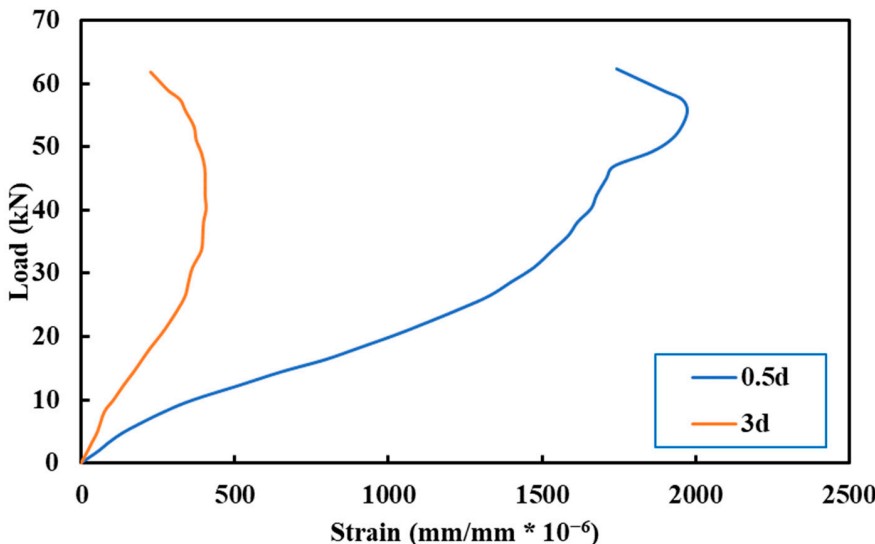

**Figure 19.** Load versus 0.5 d and 3 d concrete strains for the NCL15 slab specimen.

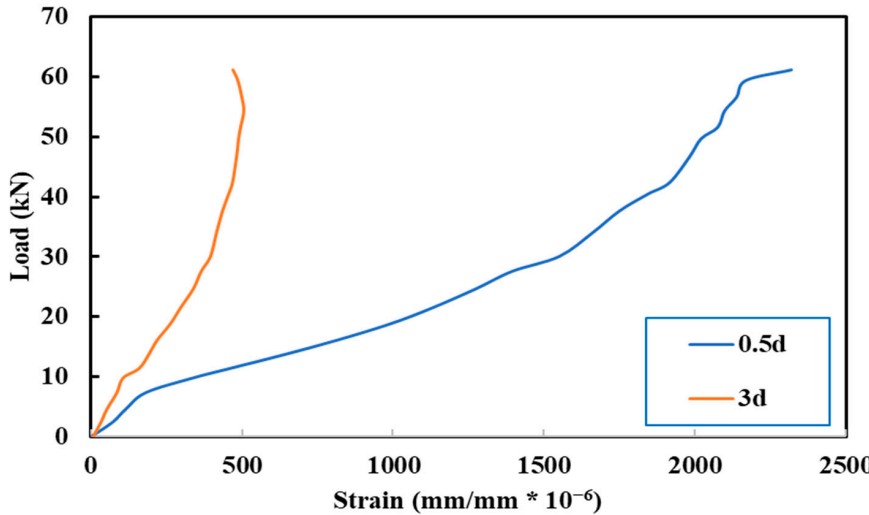

**Figure 20.** Load versus 0.5 d and 3 d concrete strains for NCL20 slab specimen.

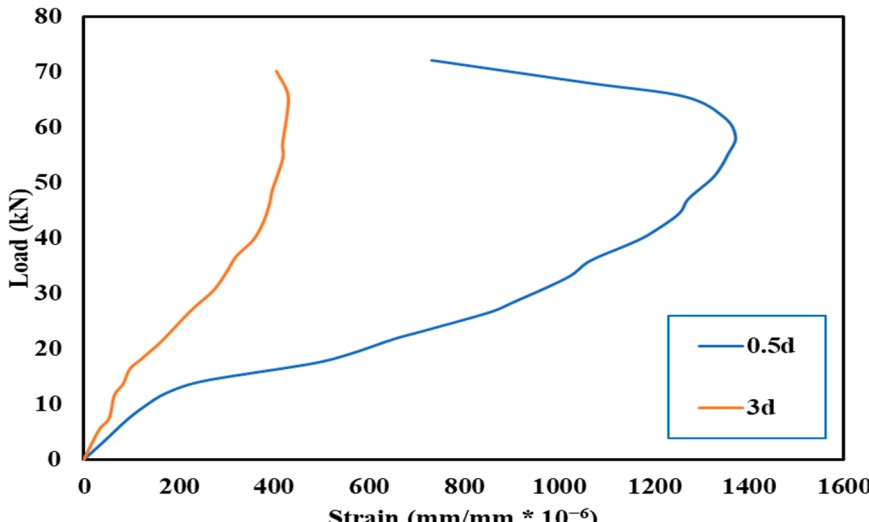

**Figure 21.** Load versus 0.5 d and 3 d concrete strains for HCL5 slab specimen.

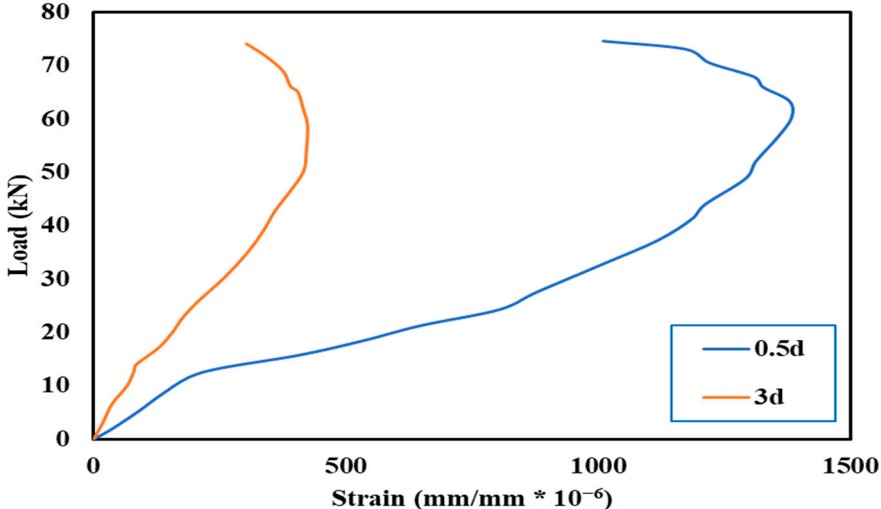

**Figure 22.** Load versus 0.5 d and 3 d concrete strains for HCL10 slab specimen.

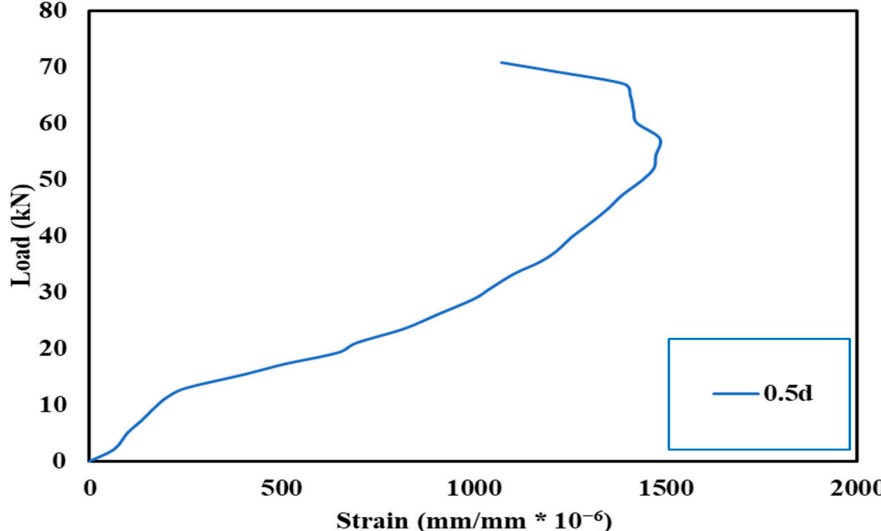

**Figure 23.** Load versus 0.5 d and 3 d concrete strains for HCL15 slab specimen.

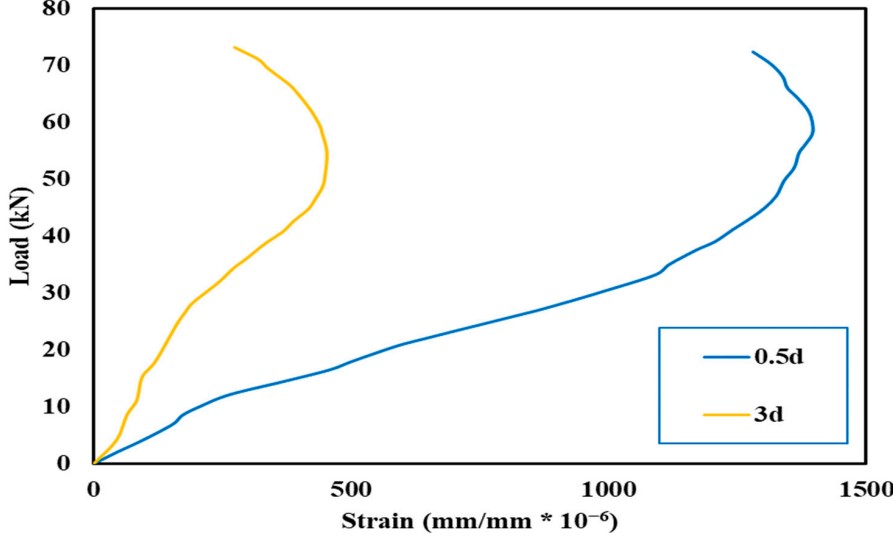

**Figure 24.** Load versus 0.5 d and 3 d concrete strains for HCL20 slab specimen.

**Table 8.** Load–concrete strains located at 0.5 d and 3 d.

| Group | Slabs | Stage | Strain at 0.5 d (Micro-Strain) | Strain at 3 d (Micro-Strain) |
|---|---|---|---|---|
| Control | N-Co | 1st crack | 376 | 110 |
| | | ultimate strain | 1754 | 392 |
| | H-Co | 1st crack | 268 | 73 |
| | | ultimate strain | 1449 | 405 |
| NCL | NCL5 | 1st crack | 359 | 103 |
| | | ultimate strain | 1448 | 367 |
| | NCL10 | 1st crack | 280 | 157 |
| | | ultimate strain | 1667 | 431 |
| | NCL15 | 1st crack | 409 | 111 |
| | | ultimate strain | 1981 | 404 |
| | NCL20 | 1st crack | 292 | 93 |
| | | ultimate strain | 2302 | 508 |
| HCL | HCL5 | 1st crack | 383 | 97 |
| | | ultimate strain | 1371 | 429 |
| | HCL10 | 1st crack | 332 | 94 |
| | | ultimate strain | 1384 | 424 |
| | HCL15 | 1st crack | 352 | not recorded * |
| | | ultimate strain | 1485 | not recorded * |
| | HCL20 | 1st crack | 371 | 91 |
| | | ultimate strain | 1395 | 453 |

* Some of the data were not recorded due to a technical problem.

### 3.9. Crack Patterns and Modes of Failures

Punching shear failure is the expected mode of failure for each flat slab. Following the first crack marking in the test, the slab specimens were subjected to continuous loading gradually and steadily until they collapsed. The test was immediately stopped at maximum loads to evaluate the cracking behavior at the tension side before the concrete took off. Figures 25 and 26 display all slab specimens' crack failure arrangement on the tension face. The initial cracking loads were observed at low loading stages because of the critical bending zone. The red color was used as an indicator for these cracks, mostly located near the column face underneath. In failed slabs, shear cracking is the most prevalent kind of crack pattern. Radial cracks in the shear perimeter began to show as the load grew and spread toward the slab supports. In the middle of the slab, other cracks appeared simultaneously. The punching zone's radial crack widths are wider than those that extend beyond the failure zone.

There was a loud sound and a sudden 10 to 30 mm penetration of the column into the slab as the slab achieved its punching load capacity. On the slabs' compression side at failure, there appeared to be no cracks except the punching cracks around the columns [30]. Slabs with irregular or circular perimeters have comparable failure patterns.

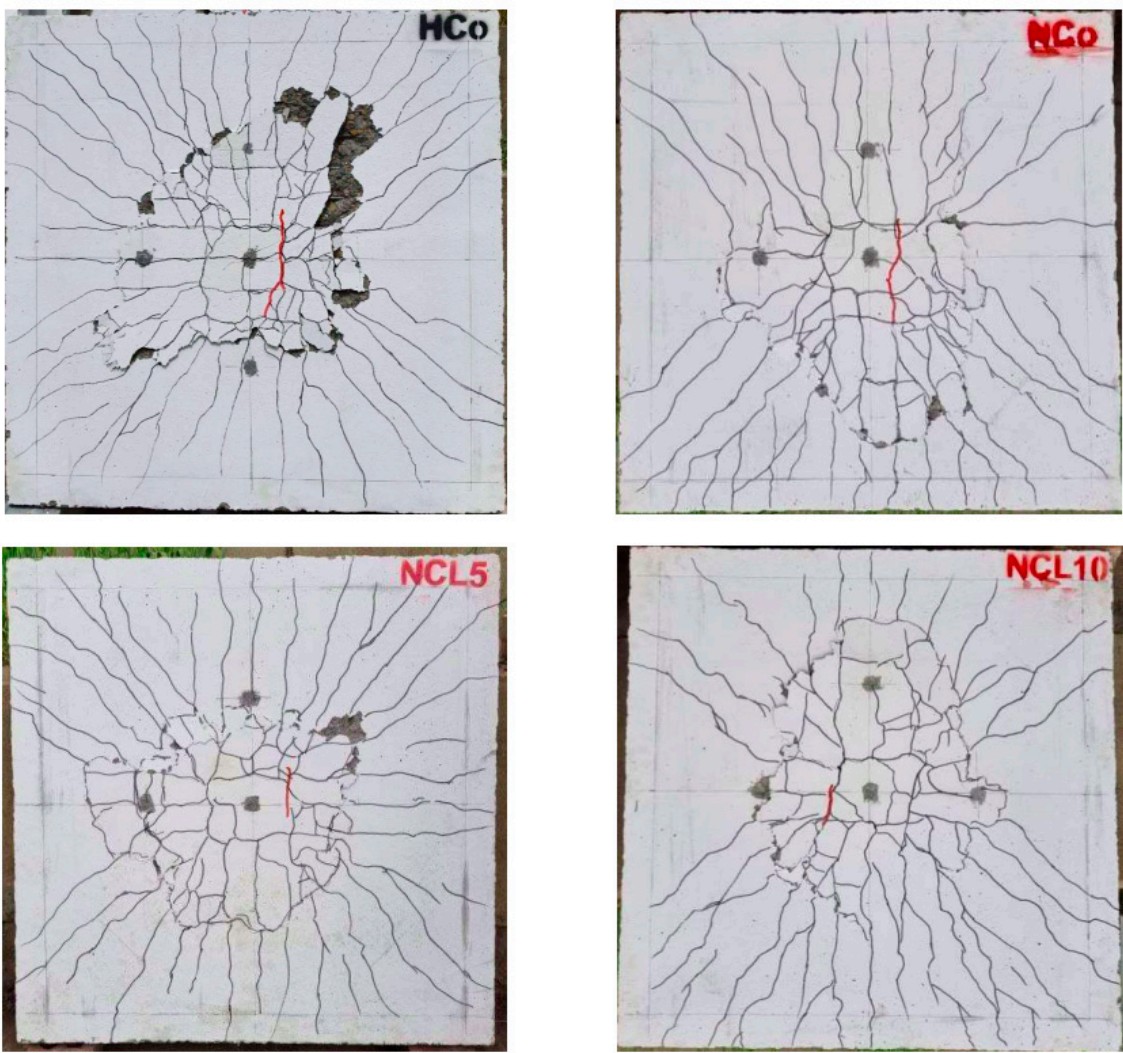

**Figure 25.** Final crack patterns in four specimens.

The punching areas in slabs N-Co and H-Co are semicircular; however, the punching areas in specimens NCL5, NCL15 and NCL20 are circular, although NCL10 is more oval-like in shape. The punching area in the slabs' HCL group is circular, except for HCL20 which is flower-like in shape. A rectangular closed crack under the column face is present at the center. In the punching areas, there are several closed cracks visible. A truncated cone is identified when reflected on the opposite surface with an enlarged area. A similar mode of failure is found in sustainable and non-sustainable normal concrete specimens; however, for higher-strength specimens, the compression failure mode is less explosive crushing in sustainable LP concrete specimens than control ones, especially at high LP ratios.

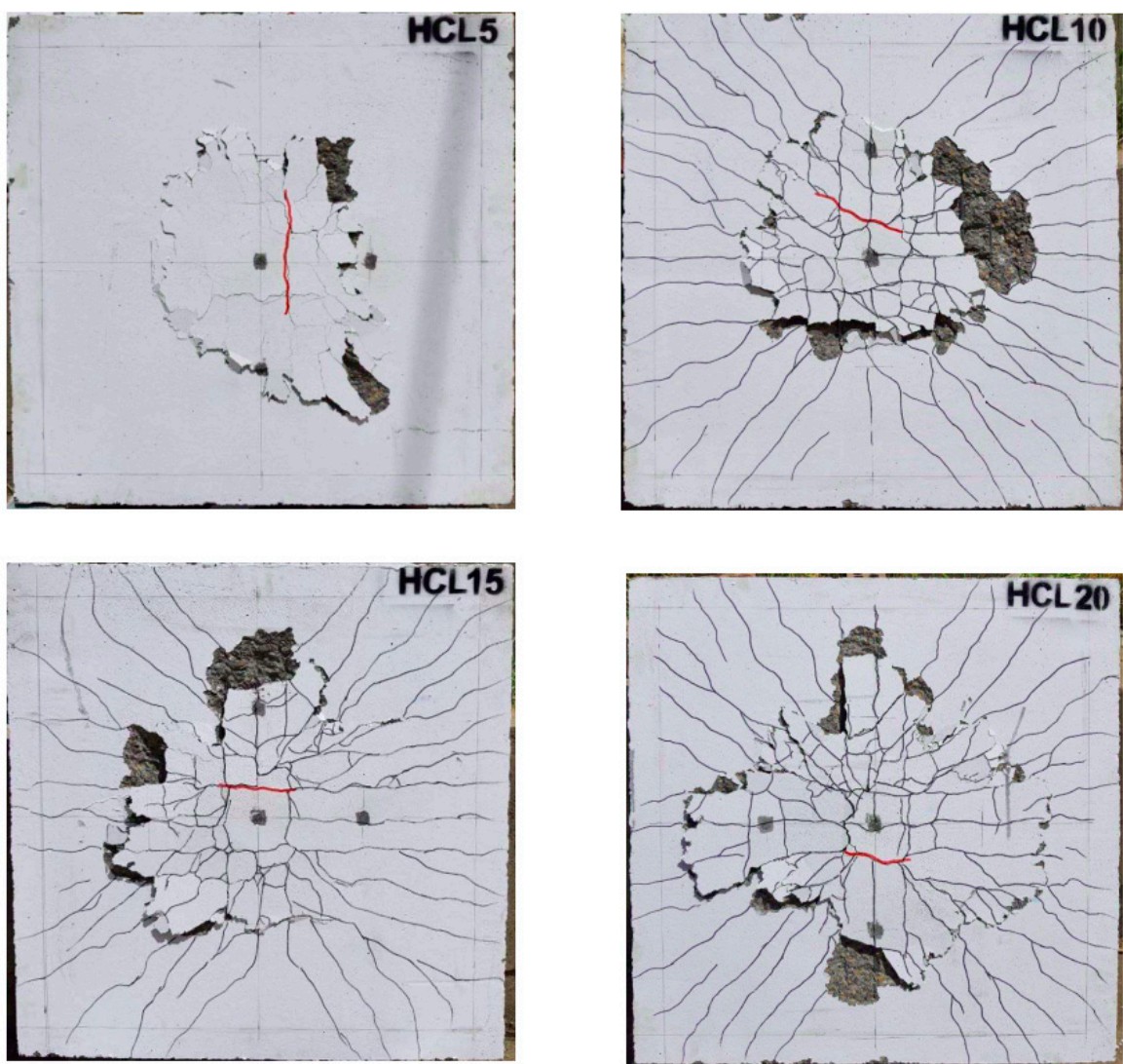

**Figure 26.** Final crack patterns in a further four specimens.

*3.10. Punching Geometrical Property*

Each slab's tension face was photographed after the slab tests. Following testing, the AutoCAD 2022 program was used to measure the punching area and perimeter of each specimen failure zone. The test results for the perimeter and punching areas of the tested slabs are displayed in Table 9. The results indicate that the tension face's punching failure zone is essentially irregular, circular, or oval. The circumference of high-strength concrete slabs is longer than that of normal-strength concrete slabs for all kinds of sustainable concrete. The perimeter for NCL slabs is between 1401 and 1621 mm, whereas for HCL slabs, it is between 1611 and 1936 mm. Generally, with the increase in the compressive strength from normal to high strength for the same replacement percentage of the slab specimens, the punching area and punching perimeter increase.

Another observational data that is computed using the punching perimeter at the slab tension side is the angle of failure, as seen in Figure 27. On each face of the column, the four angles ($\theta1$, $\theta2$, $\theta3$ and $\theta4$) of the shear failure were calculated. For each slab, the punching shear plane's average angle ($\theta$) was calculated. Table 10 represents the test results on the angle of punching failure for all slab specimens. The mean punching angles differed between 11.70° and 25.40°. In addition, the variations in the perimeters and, consequently, the distances from the column face in each direction result in various individual angles for each slab.

**Table 9.** Experimental punching area and perimeter at tension face for all slab specimens.

| Name of Slab | Group | Measured Area (mm²) | Measured Perimeter (mm) |
|---|---|---|---|
| H-Co | Control | 129,400 | 1947 |
| N-Co | | 121,000 | 1617 |
| NCL5 | NCL | 107,200 | 1503 |
| NCL10 | | 122,100 | 1621 |
| NCL15 | | 97,000 | 1573 |
| NCL20 | | 108,100 | 1401 |
| HCL5 | HCL | 112,500 | 1936 |
| HCL10 | | 100,700 | 1611 |
| HCL15 | | 137,100 | 1641 |
| HCL20 | | 195,400 | 1860 |

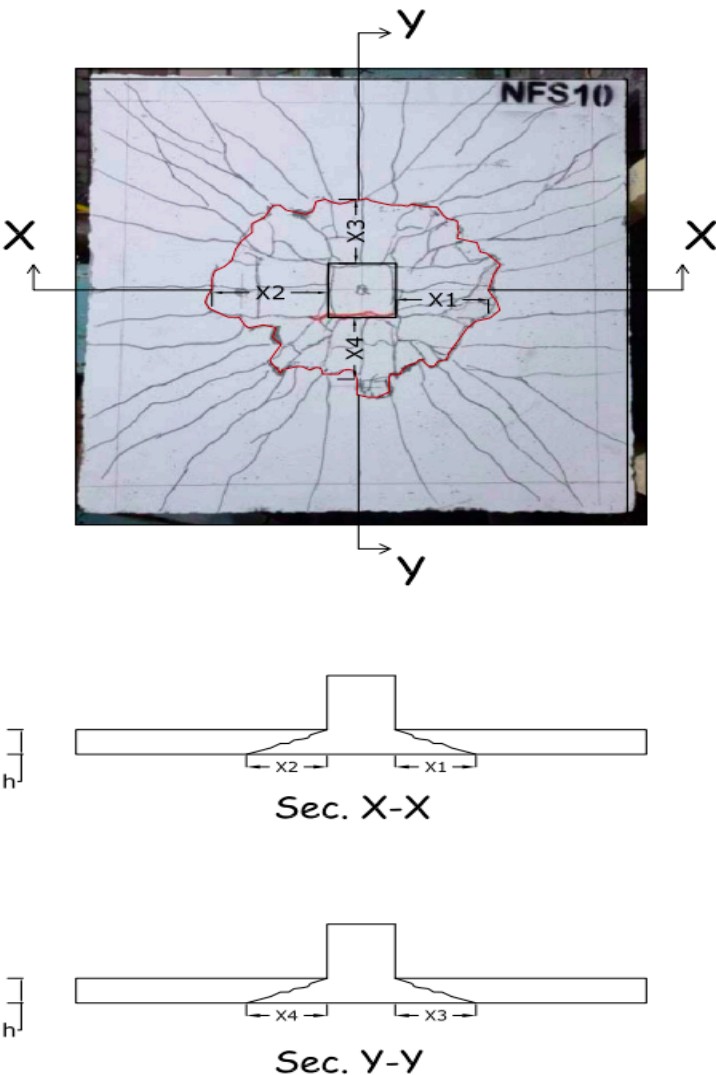

**Figure 27.** The angle of failure.

**Table 10.** Angle of punching shear for all slab specimens.

| Slab ID | h | $X_1$ (mm) | $X_2$ (mm) | $X_3$ (mm) | $X_4$ (mm) | $\theta_1$ (deg.) | $\theta_2$ (deg.) | $\theta_3$ (deg.) | $\theta_4$ (deg.) | $\theta$ (deg.) |
|---------|-----|-----|-----|-----|-----|------|------|------|------|------|
| H-Co | 50 | 108.3 | 157.4 | 167.6 | 80.7 | 24.8 | 17.6 | 16.6 | 31.8 | 22.7 |
| N-Co | 50 | 99.2 | 166.9 | 99.8 | 241.3 | 26.7 | 16.7 | 26.6 | 11.7 | 20.4 |
| NCL5 | 50 | 96.7 | 214 | 90.1 | 147.4 | 27.3 | 13.2 | 29.0 | 18.7 | 22.1 |
| NCL10 | 50 | 151.2 | 115.7 | 221.3 | 183.2 | 18.3 | 23.4 | 12.7 | 15.3 | 17.4 |
| NCL15 | 50 | 114 | 126 | 69.6 | 132 | 23.7 | 21.6 | 35.7 | 20.7 | 25.4 |
| NCL20 | 50 | 141.6 | 167.9 | 130 | 135.4 | 19.4 | 16.6 | 21.0 | 20.3 | 19.3 |
| HCL5 | 50 | 97.1 | 107.3 | 121.4 | 144.3 | 27.2 | 25.0 | 22.4 | 19.1 | 23.4 |
| HCL10 | 50 | 207.2 | 170 | 157.1 | 75.8 | 13.6 | 16.4 | 17.7 | 33.4 | 20.3 |
| HCL15 | 50 | 128 | 157.4 | 235.9 | 132.8 | 21.3 | 17.6 | 12.0 | 20.6 | 17.9 |
| HCL20 | 50 | 236.9 | 253 | 255 | 223.6 | 11.9 | 11.2 | 11.1 | 12.6 | 11.7 |

## 4. Conclusions

Based on the obtained experimental data, the following conclusions can be made:

1. As the ratio of replacing cement with LP increases, the slump of high-strength LP concrete correspondingly increases.
2. For high-strength LP concrete, the compressive strength improved with increasing LP content at 56 days. At 5% LP and 10% LP, it has the same compressive strength as the reference concrete. It ranged from 87.21 to 91.43 MPa. The percent increase in the compressive strength reached 4.5% for the concrete, including 5% LP content. However, compressive strength decreased with the increase in LP for normal-strength LP concrete. The maximum compressive strength was 35.1 MPa at 5% LP (lower than reference concrete), while the minimum compressive strength was 25.9 MPa at 20% LP, about 32.7% lower than the reference concrete.
3. In high-strength concrete, the splitting tensile strength was enhanced up to 5%. However, the splitting tensile strength decreased with the increase in LP for normal-strength concrete. The overall percentage of (splitting tensile strength/compressive strength) is 11.80% for normal strength and 6.48% for high-strength sustainable concrete. The overall percentage of splitting tensile strength/compressive strength was 11.80% for normal strength and 6.48% for high-strength sustainable concrete.
4. The use of LP up to 5% as cement replacement for normal-strength concrete did not change the elastic modulus, while for high-strength concrete, it tended to increase the elastic modulus slightly by about 8.6%. Generally, the modulus of elasticity of LP concrete was reduced as the LP increases.
5. The punching shear load capacities for normal RC slabs containing LP were higher than the control specimens up to a specific point of replacement. These increases in the punching shear load were 1.7, and 3.8% for slabs with 5, and 10% LP, respectively. However, high-strength RC slabs, compared to the control specimens, had nearly the same punching load capacity for slabs containing the same percentage replacement of LP (5 and 10%).
6. When all slabs were compared, the overall average punching strengths for LP concrete slabs with high/normal strengths were +17.3%. The corresponding values for control slabs were +18.8%. Punching ratios of LP concretes with strengths ranging from high to normal were seen to be comparable to those of the control concretes.
7. Compared to the reference slab specimen, those normal-strength RC slabs with limestone powder showed a smaller deflection at failure load, but for high-strength concrete slabs, the deflections at failure were higher than the deflection for the reference slab.

8. The average deflection value at failure load of normal strength RC slabs containing LP was 3% less than that of the reference slab. However, for high-strength concrete, the corresponding average deflection was increased by 4.7% over that of the reference slab.

9. The highest compressive strain of all specimens of normal-strength concrete slabs was obtained by slab (NCL20) at a value of 0.002302. This represents 76.7% of the ultimate strain (0.003). Slab (HCL15) exhibited the highest compressive strain among all high-strength concrete slab specimens, with a value of 0.001485, which corresponds to 49.5% of the ultimate concrete strain.

10. In general, the findings indicate that LP can be utilized to replace up to 10% of the cement in the construction of reinforced concrete flat slabs without reducing the punching load capacity (i.e., slabs with normal- or high-strength concrete). This proves that using LP concrete for structural applications is feasible as a sustainable measure.

**Author Contributions:** Conceptualization, B.K.M. and B.S.A.-N.; methodology, B.K.M.; validation, B.K.M. and B.S.A.-N.; investigation, B.K.M.; resources, B.K.M.; writing—original draft preparation, B.K.M.; writing—review and editing, B.K.M. and B.S.A.-N.; supervision, B.S.A.-N.; project administration, B.S.A.-N. All authors have read and agreed to the published version of the manuscript.

**Funding:** This research received no external funding.

**Institutional Review Board Statement:** Not applicable.

**Informed Consent Statement:** Not applicable.

**Data Availability Statement:** Data will be made available on request.

**Acknowledgments:** The authors appreciate the support received from the College of Engineering, University of Sulaimani for providing laboratory facilities.

**Conflicts of Interest:** The authors declare that they have no known competing financial interests or personal relationships that could have appeared to influence the work reported in this paper.

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
