# Peer review of "Effectiveness of Limestone Powder as a Partial Replacement of Cement on the Punching Shear Behavior of Normal- and High-Strength Concrete Flat Slabs"

_sustainability, doi:10.3390/su16052151_

Round 1
Reviewer 1 Report
Comments and Suggestions for Authors
1、 It is recommended that the author describe the meaning of RC in the abstract for readers' understanding.
2、 There are four types of cement in lines 68 and 69.
3、 It is recommended that the author use less spoken expressions in paragraph 6 of the introduction.
4、 The subheading after the second heading is misnumbered.
5、 It is recommended that the author explain the LVDT on line 191 for the reader's understanding.
6、 It is recommended to write all the picture names at the bottom of the picture.
7、 It is recommended to add a picture between Figures 14 and 23 of the installation position of the strain gauge.
8、 In line 400, the literature or normative sources of the stated knowledge points should be stated.
9、 The description of the experimental phenomenon reflected in Figure 24 is too complicated, so it is recommended to use concise language.
10、 In Figure 25, it is suggested to label the angle of failure.
Comments on the Quality of English Language1、 It is recommended that the author use less spoken expressions in paragraph 6 of the introduction.
2、 The description of the experimental phenomenon reflected in Figure 24 is too complicated, so it is recommended to use concise language.
Reviewer 2 Report
Comments and Suggestions for Authors 1. The Introduction section is written very poorly: - Repetition of the same thoughts, but in different words; - Paragraphs are not logically connected to each other; - The text contains long and difficult to understand sentences, for example: L39-43 "One of the many ways that engineers can implement sustainability into their work is through the materials that they use, the current conventional materials that are used in construction have a limited availability and they also create large carbon footprints, meaning that the procedures that are required to make, transport, install and dispose of them are all very damaging to the environment and require the use of large amounts of fossil fuels and other natural resources that are depleting in availability". We recommend dividing this sentence into two simpler sentences as follows: One of the many ways that engineers can implement sustainability into their work is through the materials that they use. The current conventional materials that are used in construction have a limited availability and they also create large carbon footprints, meaning that the procedures that are required to make, transport, install and dispose of them are all very damaging to the environment and require the use of large amounts of fossil fuels and other natural resources that are depleting in availability"; - There is no clear statement of the purpose of the study. Overall, it is recommended that this section be completely rewritten. 2. Very low quality of article design. General conclusion: the article is written in poor English, which is very difficult to understand the authors' ideas and the results obtained, so the manuscript requires serious revision.Comments on the Quality of English Language
1. The Introduction section is written very poorly: - Repetition of the same thoughts, but in different words; - Paragraphs are not logically connected to each other; - The text contains long and difficult to understand sentences, for example: L39-43 "One of the many ways that engineers can implement sustainability into their work is through the materials that they use, the current conventional materials that are used in construction have a limited availability and they also create large carbon footprints, meaning that the procedures that are required to make, transport, install and dispose of them are all very damaging to the environment and require the use of large amounts of fossil fuels and other natural resources that are depleting in availability". We recommend dividing this sentence into two simpler sentences as follows: One of the many ways that engineers can implement sustainability into their work is through the materials that they use. The current conventional materials that are used in construction have a limited availability and they also create large carbon footprints, meaning that the procedures that are required to make, transport, install and dispose of them are all very damaging to the environment and require the use of large amounts of fossil fuels and other natural resources that are depleting in availability"; - There is no clear statement of the purpose of the study. Overall, it is recommended that this section be completely rewritten. 2. Very low quality of article design. General conclusion: the article is written in poor English, which is very difficult to understand the authors' ideas and the results obtained, so the manuscript requires serious revision.
Reviewer 3 Report
Comments and Suggestions for Authors
The conducted complex scientific research and the presented results have undoubted relevance and practical significance. However, after reading the article, several recommendations arose, taking into account which the authors will be able to convey the significance of their work to the readers to a greater extent:
1. The authors should thoroughly revise the text in terms of content and grammar: What does 243 mean on line 192? On line 197 the reference to Table 6, on line 198 the reference to Table 6 is repeated, but it is obviously referring to Table 5. "In Table 6 illustrates" on line 211 appears to be a grammatical error. The axis caption of Figure 5 contains a typo in the word "Replacement". What is "Relation-ship" on line 305? I'm afraid there are too many of these kinds of errors and typos.
2. The procedure by which Figures 14 - 23 are obtained should be more fully described and illustrated. It is difficult to imagine how the strain can begin to relax with a continued increase in load. Reference should be made to standard methodology.
3. In some sections, the authors present results, but there is no discussion and analysis with references. In its current form, the paper looks like a research report and needs some discussion of the results obtained.
Comments on the Quality of English Language-
Round 2
Reviewer 2 Report
Comments and Suggestions for Authors
The authors made the necessary changes to the text of the manuscript. The article may be accepted for consideration.